# The Dynamics of Water Wells Efficiency Reduction and Ageing Process Compensation

**Krzysztof Polak** [1,*] , **Kamil Górecki** [2] **and Karolina Kaznowska-Opala** [3]

1   Centre of Energy, Faculty of Mining and Geoengineering, AGH University of Science and Technology, 30059 Krakow, Poland
2   WellsCtrl.Tech, 30059 Krakow, Poland; kamil.gorecki85@gmail.com
3   Department of Surface Mining, Faculty of Mining and Geoengineering, AGH University of Science and Technology, 30059 Krakow, Poland; kazn@agh.edu.pl
*   Correspondence: kpolak@agh.edu.pl

**Abstract:** Water wells play an increasingly important role in providing water for the civilian population all over the world. Like other engineering structures, wells are subject to ageing processes resulting in degradation, which is observed as a reduction in hydraulic efficiency throughout their lifespan. To date, it has been found that the ageing process of a well is determined by a number of factors. The mathematical description of this process can be simplified. Drawing on Jacob's equation, this paper presents the course of the degradation process as a variable depending on operation time, well loss and flow rate. To apply the determined relationships in practice, simplifying assumptions were adopted, which make it possible to determine the moment of ageing compensations of the degradation processes. It was also demonstrated that the degradation process may be slowed down by the appropriate selection of initial operating parameters. The presented discussion highlights the significance of parameters $\alpha$, $\delta$ and exponent $\beta$. The relation between hydraulic resistances in an aquifer and in the engineering structure is closely connected with these values. The presented arguments indicate that step drawdown tests provide the necessary information which allows tracking changes in the ageing processes occurring in the engineering structure. The analysis of the drawdown test results makes it possible to determine the moment when the necessary adjustments in the operating parameters of a water well should be performed. Eventually, it allows maintaining the high hydraulic efficiency of the intake and extending the lifespan of the well in accordance with the principle of sustainability.

**Keywords:** water well; hydraulic efficiency; degradation; engineering structure; well ageing; lifespan; well operation; water well management; sustainable efficiency

## 1. Introduction

Groundwater extraction is still growing all over the world. It is estimated that within half a century, i.e., between 1960 and 2010, groundwater use doubled [1]. The drilling of new groundwater wells is required. However, the number of boreholes is increasing not only due to population growth, but also due to a decrease in productivity of the existing assets. As well as other technical facilities wells are undergoing the process of degradation. As with any other technical facility, the current technical condition of a drilled well can be described with an efficiency index. Well efficiency is defined by Rorabaugh as the ratio of the theoretical drawdown computed by assuming that no turbulence is present to the drawdown in the well [2]. Walton defines the efficiency of a well as the ratio of the theoretical specific capacity to the actual specific capacity of the well [3]. Bierschenk concludes that well efficiency may be defined as the ratio of the theoretical drawdown to the measured drawdown inside

the well. He also presents multiple examples of efficiency curves calculated using step-drawdown tests carried out for wells located in the Middle East [4]. Therefore, well loss determines the hydraulic efficiency of a wellbore.

Well loss can be evaluated using the step-drawdown test. In Jacob's equation, well loss is described as the non-linear (non-laminar) flow regime component $CQ^2$ [5]. In some cases, changes of the drawdown can be explained by the Rorabaugh formula, where the power exponent $p$ in the expression $CQ$ is different from 2. Mackie, as cited by Atkinson et al. [6], reviewed the results of more than 20 carefully conducted step-drawdown tests of wells completed in fractured rock aquifers and concluded that most of the responses fell into one of the three categories where specific drawdown versus discharge rate is: linear, polynomial with power exponent equal to 2, or different from 2. The results of a study by Motyka and Wilk regarding the determination of the non-linear flow zone around several dozen wells drilled in fractured rocks indicate that the radius of turbulent flow zones are usually from 0.5 to 5.0 m, although in most cases they do not exceed 1 m [7].

Atkinson et al. concludes that high pumping rate moves the turbulent flow regime into the fissured aquifer, which is the reason for power exponent changes in Rorabaugh formula [8]. Klich et al. considered that the nature of the drawdown versus well-discharge curve depends on the range of well discharge [9]. Shekhar carried out the series of filed test for unconsolidated soils. He noticed that flow into the well screen is of a turbulent nature even for low discharge. The decrease in efficiency with increase of discharge can be considered from the perspective of increase in turbulence on account of the increase in discharge. As the drawdown increases the curvature of flow path increases leading to greater head loss. Therefore, well efficiency calculated can be regarded as a reflection of head loss on account of the laminar flow from the aquifer [10].

Much of the literature on the step-drawdown test focuses on the methods of determining aquifer resistance coefficient for linear water flow $B$, well resistance coefficient for non-linear flow $C$ and $p$ (if assumed not to be 2) parameters. Several methods are compiled by Kruseman and de Ridder [11]. One of the simplest graphical approaches is attributed to Jacob [5], Bruin and Hudson [12], Bierschenk and Wilson [13]. The parameters of the step-drawdown test can also be determined numerically. Miller and Webber [14] present an iterative method for solving the equation, while Labadie and Helweg developed a computer program for step-drawdown test data interpolation with the FASTEP procedure [15]. Avci proposes a method of analysis which calculates the aquifer and well loss from best fitting the log-log relationships of the difference of specific capacity and superposition of incremental pumping to the flow rate [16]. Wen et al. built the analytical model for all the types of curves considered for water well in confined aquifer. They noticed that all the curves at the wellbore approach the same asymptotic straight line in log s–log t scales [17]. Determining hydraulic characteristics of production wells from step-drawdown test data were also proposed by Jha et al. The characteristics of production wells such as aquifer loss coefficient, well loss coefficient, well specific capacity and well efficiency were determined by both the Genetic Algorithm optimization technique and the widely used graphical method. The developed computer programs also provide information about the condition of production wells and facilitate the construction of well characteristic curves [18].

Helweg proposes using General Well Function (GWF) to interpret the step-drawdown test. It allows more reliable results to be achieved for situations when the time that wells are continuously pumped varies greatly [19]. Kawecki presents a method for calculating total well loss based on the step drawdown test. In this method, the total loss can be estimated as a function of discharge rate. However, the real range radius of the borehole and also the specific storage of the confined aquifer are changeable at different stages of the step drawdown test [20]. An analytical solution for the constant pumping test in fissured porous media is presented by de Smedt. The solution is based on the dual-porosity approach with pseudo-steady state exchange between fissures and matrix. Proposed solution provides approach to analyze pumping test in fissured porous media characterized by double-porosity effects [21].

Singh proposes variable pumping replacing the step drawdown test, which does not require steady-state conditions at each step. This method allows the simultaneous estimation of both aquifer

and well loss parameters [22]. The same author proposes a method for identifying head-loss from early drawdowns, where well loss could be evaluated from a short-duration pumping test in transient conditions. However, this method requires measurements in the pumped well and observation wells [23]. In contrast to this, Avci et al. proposed an analysis technique for interpreting transient step-drawdown test data. The method is based on taking the derivative of the drawdown with respect to time for the entire pumping test period, eliminating the constant well-loss terms. The conducted tests showed that the method allows the generation of the aquifer function $B(t)$ for the entire duration of the step-drawdown test [24]. The choice of the model is the most difficult choice that the analyst of such a hydraulic test has to make since a wrong model can only lead to the wrong conclusions and failure of the borehole [25]. Although the physical model is well-known and widely used, there is a strong need to improve the techniques for estimating uncertainty associated with parameters derived from the interpretation of well tests [26].

The results of the step drawdown test could be applied to hydrogeology research. Dufresne presents the data used to develop a regional groundwater model which facilitates water planning and sustainability [27]. However, the step drawdown test is the most common tool to assess well performance and the hydraulic statement of the clogging. Many types of clogging processes are described in the book by Houben and Treskatis [28]. It involves a detailed discussion of the causes and effects of physical, chemical, electrochemical, and biochemical clogging. It also describes the methods and possibilities of rehabilitation of the well screen, gravel-pack and well-tube.

Van Beek presents the rehabilitation of mechanically clogged discharge wells. The research conducted so far has shown that the increase in flow velocity around a wellbore mobilizes fine particles, which are then transported through the porous structure. This may lead to the clogging of slots of the screen, thus reducing both porosity and permeability [29]. The mechanical clogging process in unconsolidated aquifers near the water supply wells is presented by de Zwart [30] and de Zwart et al. [31]. Blackwell et al. analyses particulate damage, which is often cited as a cause of permeability impairment around boreholes, resulting in a decline in well performance. Particulate movement and redistribution under borehole operating conditions have been assessed for a range of artificial and natural formations. Particulate damage cannot be effectively eliminated using normal development and rehabilitation techniques. This movement may occur at different stages in the life of the borehole, i.e., drilling, development and operational damage. Therefore, the operating regime has an important effect on well performance, which should be regularly monitored. If deterioration reaches approximately 10%, rehabilitation action can be initiated. Allowing deterioration to exceed 25% can greatly increase the cost of successful rehabilitation [32].

The example of chemical clogging including iron and iron-reducing bacteria is also presented by Hitchon [33]. The clogging of deep infiltration wells was also presented by de la Loma Gonzales, where several different types of chemical clogging and rehabilitation methods were compared and evaluated using a specific case of a well system [34]. The requirements for the economical construction and operation of drilled wells was also described by Treskatis et al [35]. These authors propose preventive maintenance for well rehabilitation as an essential guard against the development of non-soluble incrustations of Fe- and Mn-(hydr)oxides by back wash procedures or high-velocity, horizontal jetting techniques at an early stage of biochemical development aid to keep well efficiency at a high level. They also point out that surveillance of well efficiency and changes of drawdown can help to reduce the effort of chemicals for well rehabilitation. Van Beek concludes that well bore clogging may be prevented by regular intermittent abstraction. During rest, as there is no incoming flow, the particle accumulations will disintegrate. However, there is no clear picture of this disintegration process, an objective criterion for the rest time is lacking. It might be possible to shorten the rest period by reversing the flow direction with the abstraction flow velocity at least. Well bore clogging may also be counteracted by proper well development. Currently the criterion for a developed well is the absence of sand in the abstracted water. Actually, well development should be maximal until there is no further increase in specific capacity [36].

Houben et al. conclude that dramatic increases in head losses occur when clogging has reduced the effective porosity of the gravel pack by ~65%, the open area of the screen by ≥98%, and the casing diameter by ~50%. However, the clogging of the gravel pack is the most important object influencing well ageing. Moreover, obstructions in the gravel pack are much more difficult to remove. So, regular monitoring of well performance is needed, since processes in the gravel pack are difficult to track directly [37]. Therefore, low well efficiency contributes to the increased operating costs of wells. Helweg and Bengstone considered the economic approach the most profitable pumping rate in the whole life-span of the well, including of substitute well drilling [38]. Cost optimization throughout the life-span of a well is also the subject of a paper by Hurynowicz and Syczewa, who conclude that maintaining relatively low operating costs of a well depends on its appropriate design and operating conditions over several dozen years of its operation [39].

To sum up the above, well ageing causes well loss which is time-dependent. However, well loss is a naturally occurring phenomenon that depends on the hydraulic features of the aquifer, well construction, hydrogeochemistry, borehole drilling, well development and operation. The literature review presented above demonstrates that step-drawdown tests are typically used to estimate the hydrogeological parameters, well loss and well efficiency. Step-drawdown test results are also used to assess the current state and the effects of rehabilitation operations.

## 2. Materials and Methods

Water supply wells should be characterized by stable operation over a long period of time. They should therefore be operated with the highest possible hydraulic efficiency. After by van Tonder et al. [25], the rate at which a borehole can be pumped without lowering the water level below a set level is called "sustainable yield". Piscoppo and Summa discussed an experiment to estimate the pumping rate to ensure that the sustainable yield concept is realized [40]. At the beginning a step drawdown test was carried out, then the efficiency of the well was determined. Based on the results of the tests performed, the constant-head for the functioning of the well was defined. It was calculated that for a drawdown of around 12 m, the well efficiency is greater than 75%. The well was equipped with suitable submersible pump unit, then it was monitored on-line by one year. The pumping rate was defined as the discharge rate that will not cause the water level in the well to drop below a prescribed limit. In conclusions, authors emphasize that in some cases the constant-head pumping can be an alternative method of well management. Analysis of the recorded data changes shown that the discharge rate of the well trend with time was similar to that of the springs' hydrograms. Moreover, the water volume extracted did not exceed the recharge [40]. Subsequently, constant head model was compared with a constant flow rate model by Baiocchi et al. Based on numerical modelling, the authors concluded that constant flow rate model could be particularly useful when the problem is one of determining the sustainable yield of a single well from aquifers with low hydraulic diffusivity and when an extensive monitoring of the aquifer is not economically viable [41]. Unfortunately, the researchers did not evaluate whether both models had an impact on the well efficiency at the end of the research period.

In contrast to examples mentioned above, the authors of this paper have developed a concept of well operation while maintaining its constant efficiency, which could be also called "sustainable efficiency". This concept required the adoption of certain initial assumptions and was then determined theoretically using formulas evaluating the hydraulic efficiency of the well, taking into account the elements of the optimization calculus and also the ageing function of technical objects. Literature on hydrogeological issues states that the efficiency of a well can be described using an expression involving a rational function in the form proposed by Bierschenk [2]:

$$\eta = \frac{s_1}{s} = \frac{s_1}{s_1+s_2} = \frac{BQ}{BQ+CQ^2} = \frac{1}{1+\frac{C}{B}Q}$$
$$\eta = \frac{1}{1+\alpha Q}, \, for \, \alpha = \frac{C}{B} \tag{1}$$

where:

$\eta$ hydraulic efficiency [-];
$s_1$ aquifer loss [L];
$s$ drawdown observed in the pumping well [L];
$s_2$ well loss in the well-screen adjacent zone [L];
$Q$ volumetric flow rate [$L^3/T$];
$B$ aquifer resistance coefficient for laminar water flow [$T/L^2$];
$C$ well resistance coefficient for turbulent flow [$T^2/L^5$];
$\alpha$ parameter describing the prevalence of turbulent flow [$T/L^3$].

This concept is presented in a schematic Figure 1.

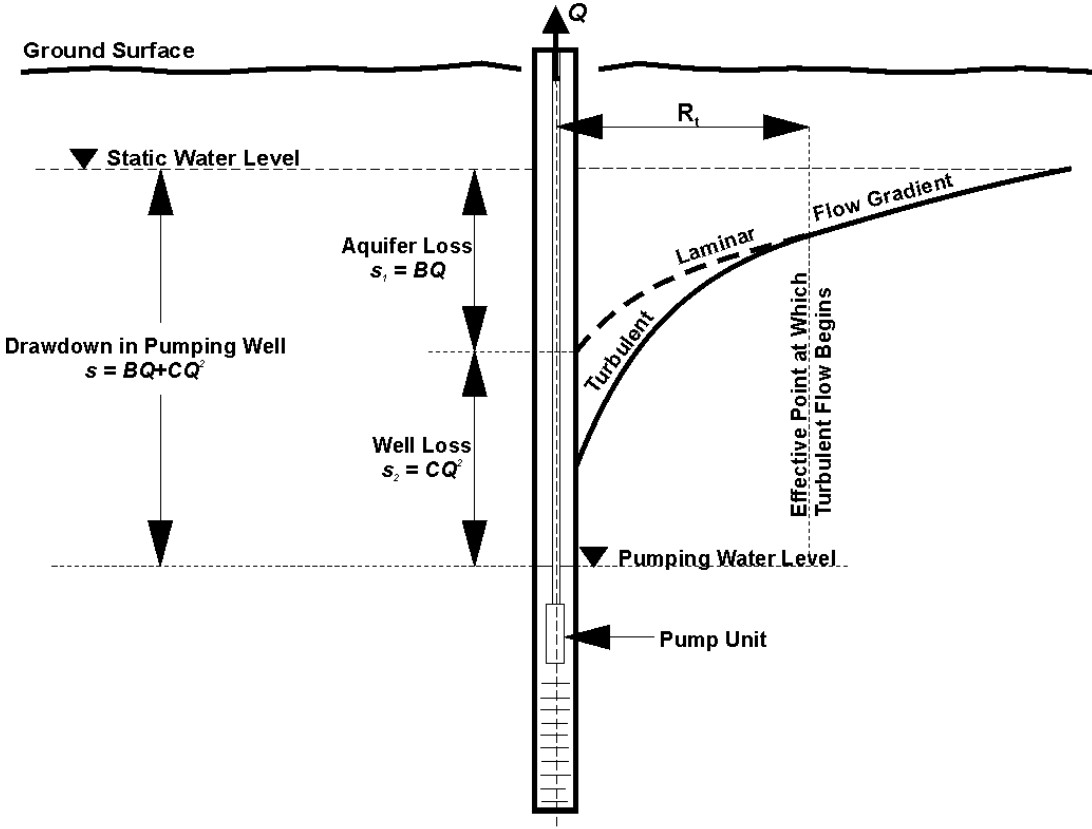

**Figure 1.** Head losses in pumped well [4]; where: $R_t$—distance from center of well to effective point formation where transition from laminar to turbulent flow takes place.

At the beginning of the consideration presented below, it was assumed that the state of knowledge is sufficient to be used to select the optimum operating parameters of new wells and also to perform ongoing adjustments to counteract the intensive ageing of operated wells. So far, this subject matter has not been discussed in detail in the literature. For this reason, the authors present the results of their own research and considerations regarding the dynamics of the ageing process and the possibilities of reducing its rate. Below the article presents a process of degradation of a water well as an engineering structure. As previously mentioned, well efficiency function is time dependent. It means that a degradation process can be described as efficiency function in time. Therefore, well ageing is derivative from well efficiency in time.

For this purpose, the initial assumptions relating to the ageing process of the well could be defined by taking into account the state of knowledge presented in previous section:

**Assumption 1.** *Aquifer resistances are slow-variable. This assumption is valid for water supply wells where hydrogeological conditions do not change over time. In relation to drainage wells, the condition is usually not applicable due to the lowering of the water level and the relatively short life-span of the wells.*

**Assumption 2.** *The value of well loss ($s_2$) should not significantly exceed the value of aquifer loss ($s_1$). This assumption is usually relevant to new wells where the well-loss is insignificant in comparison to total drawdown. However, such criteria is also applicable to other wells in good hydraulic condition.*

Assumption 1 assumes that aquifer resistances are slow variables. This means that $\dot{B}(t) \approx 0$ in comparison to the rates of changes in the well resistance parameters ($C$) or its discharge ($Q$)—$\dot{B}(t) \ll \dot{C}(t)$ ($or$ $\dot{Q}(t)$) (accurate to the consistency factor of the dimension of a physical quantity). Therefore, if the said coefficients depend on time in a way that: $C = f(t)$, $B = const(t)$, $\alpha = \frac{C}{B} = g(t)$ $and$ $Q = h(t)$, then well efficiency will also be a function of time $\eta = \eta(t)$. The well efficiency function can be expanded into a Taylor series around moment $t = t_0$ as:

$$\eta(t) = \eta(t_0) + \sum_{n=1}^{k} \frac{1}{n!} \frac{d^{(n)}\eta}{dt^{(n)}}\bigg|_{t=t_0} (t-t_0)^n \tag{2}$$

or by writing the above equation as follows:

$$\eta(t) = \eta(t_0) + \frac{1}{1!}\dot{\eta}(t_0)\cdot(t-t_0) + \frac{1}{2!}\ddot{\eta}(t_0)\cdot(t-t_0)^2 + \frac{1}{3!}\dddot{\eta}(t_0)\cdot(t-t_0)^3 + \ldots \tag{3}$$

Increases in the relevant variables may be used here—the increase in the independent variable (time) $\Delta t = t - t_0$ and in the dependent variable (well efficiency) $\Delta \eta = \eta - \eta_0$. Then, the following relationship is present:

$$\Delta\eta(t) = \dot{\eta}(t_0)\cdot\Delta t + \frac{1}{2}\ddot{\eta}(t_0)\cdot\Delta t^2 + \frac{1}{6}\dddot{\eta}(t_0)\cdot\Delta t^3 + \ldots \tag{4}$$

The time derivative of well efficiency is:

$$\dot{\eta}(t) = \frac{d}{dt}\left(\frac{1}{1+\alpha Q}\right) = \frac{-1}{(1+\alpha Q)^2}(\dot{\alpha}Q + \alpha\dot{Q})\dot{\eta}(t) = -(\dot{\alpha}Q + \alpha\dot{Q})\cdot\eta^2 \tag{5}$$

The second-order time derivative can be described by the expression:

$$\ddot{\eta}(t) = \frac{d}{dt}(\dot{\eta}) = (-1)\frac{(\dot{\alpha}Q+\alpha\dot{Q})'\cdot(1+\alpha Q)^2 - [(1+Q)^2]'\cdot(\dot{\alpha}Q+\alpha\dot{Q})}{[(1+\alpha Q)^2]^2}$$

$$\ddot{\eta}(t) = (-1)\frac{(\ddot{\alpha}Q+\dot{\alpha}\dot{Q}+\dot{\alpha}\dot{Q}+\alpha\ddot{Q})\cdot(1+\alpha Q)^2 - 2(1+\alpha Q)\cdot(\dot{\alpha}Q+\alpha\dot{Q})\cdot(\dot{\alpha}Q+\alpha\dot{Q})}{(1+\alpha Q)^4}$$

$$\ddot{\eta}(t) = (-1)\frac{(\ddot{\alpha}Q+2\dot{\alpha}\dot{Q}+\alpha\ddot{Q})\cdot(1+\alpha Q)^2 - 2(1+\alpha Q)\cdot(\dot{\alpha}Q+\alpha\dot{Q})^2}{(1+\alpha Q)^4} \tag{6}$$

$$\ddot{\eta}(t) = (-1)\frac{[(\ddot{\alpha}Q+2\dot{\alpha}\dot{Q}+\alpha\ddot{Q})\cdot(1+\alpha Q) - 2(\dot{\alpha}Q+\alpha\dot{Q})^2](1+\alpha Q)}{(1+\alpha Q)^4}$$

$$\ddot{\eta}(t) = \frac{2(\dot{\alpha}Q+\alpha\dot{Q})^2 - (\ddot{\alpha}Q+2\dot{\alpha}\dot{Q}+\alpha\ddot{Q})\cdot(1+\alpha Q)}{(1+\alpha Q)^3}$$

The above can be expressed in a simplified manner as:

$$\ddot{\eta}(t) = \frac{2}{\eta}(\dot{\eta})^2 - \eta^2(\ddot{\alpha}Q + 2\dot{\alpha}\dot{Q} + \alpha\ddot{Q})$$

$$\ddot{\eta}(t) = \frac{2}{\eta}(\dot{\eta})^2 - \eta^2\frac{d^2}{dt^2}(\alpha Q) \tag{7}$$

In Assumption 2 the value of well loss ($s_2$) should not significantly exceed the value of aquifer loss ($s_1$). This means that:

$$\frac{s_2}{s_1} \ll 1$$
$$\frac{s_2}{s_1} = \frac{CQ^2}{BQ} = \frac{C}{B}Q = \alpha Q \ll 1 \tag{8}$$

Therefore, the rate of change is small in comparison with other elements in the expression for the second-order time derivative of well efficiency:

$$\frac{d^2}{dt^2}\left(\frac{s_2}{s_1}\right) = \frac{d^2}{dt^2}(\alpha Q) \approx 0$$
$$\ddot{\eta}(t) = \frac{2}{\eta}(\dot{\eta})^2 - \eta^2 \frac{d^2}{dt^2}(\alpha Q) \approx \frac{2}{\eta}(\dot{\eta})^2 \tag{9}$$
$$\ddot{\eta}(t) \approx \frac{2}{\eta}(\dot{\eta})^2$$

Including in the expression the first-order derivative in the form: $\dot{\eta}(t) = -(\dot{\alpha}Q + \alpha\dot{Q})\cdot\eta^2$ provides the clear form:

$$\ddot{\eta}(t) = \frac{2}{\eta}\left[(-1)(\dot{\alpha}Q + \alpha\dot{Q})\cdot\eta^2\right]^2$$
$$\ddot{\eta}(t) = 2(\dot{\alpha}Q + \alpha\dot{Q})^2\cdot\eta^3 \tag{10}$$

The above considerations can be simplified by applying approximations in the solution, in which higher-order terms are negligible.

## 3. Results

The above equations describe the function of well degradation as a derivative of efficiency in time. However, it can be used to determinate the rate of ageing as well as to counteract this process. Issues can be considered in a simplified form as first-order approximation (kinetic), or in expansions of second-order approximations (dynamic). The higher-order terms, starting from the third order may be considered negligible. Changes in the function of the state of the well are presented in two independent variants, applying the principle of caeteris paribus. The first variant describes the ageing of the object and the decrease in hydraulic efficiency. The second variant describes the recovery of the lost efficiency of the well by changing the operational parameters of the well.

### 3.1. First-Order Approximation

In this case, higher-order terms are negligible, starting from the second order. In such a case the expression expanding the Taylor series for a function to a power series is identical to the expression for the differential of the function (adjacent to point ($t_0$; $\eta(t_0)$)):

$$\Delta\eta(t) \approx \dot{\eta}(t_0)\cdot\Delta t \tag{11}$$

which, after inserting the time derivative value for moment $t_0$ equals:

$$\Delta\eta = -(\dot{\alpha}Q + \alpha\dot{Q})\cdot\eta_0^2\cdot\Delta t \tag{12}$$

To ensure clarity, subscripts are omitted for derivatives of functions $Q$ and $\alpha$ at the moment of time $t_0$. However, this fact should be taken into account when performing numerical calculations.

Time ranges can be matched in such a way that only one of the parameters, $\alpha$ or $Q$, will change distinctively over time, which would allow it to provide a significant contribution to the expression for changes in well efficiency throughout its lifespan. This will be discussed below on two extreme cases, which are viable on an engineering scale.

*Case 1–Well Ageing*

In this case it was assumed that the flow rate is invariable or slow-variable, while well resistances in relation to aquifer resistances increase considerably. This causes the deepening of the dynamic water table while maintaining a constant flow rate. This leads to a considerable reduction in the hydraulic

efficiency of the well. Therefore, the turbulent flow of the reservoir fluids migrating in the near well zone acquires a high significance with time. This is connected with well ageing, which is a result of sediment accumulation in the functional part of the screen, clogging of the porous or fissure area of the medium directly adjacent to the screen and even the appearance of particulates inside the screen of the pumping well.

In other words: $\dot{Q} \approx 0$ *and* $\dot{\alpha} > 0$. Therefore:

$$\Delta\eta_{age} = -(\dot{\alpha}Q + \alpha\dot{Q})\cdot\eta_0^2\cdot\Delta t_{age}\Delta\eta_{age} = -\dot{\alpha}Q\cdot\eta_0^2\cdot\Delta t_{age} \tag{13}$$

As can be easily determined, the inequality $\Delta\eta_{age} < 0$ over time will always be satisfied.

*Case 2—Compensation of Ageing*

In this case the flow rate decreases with time and the well resistances do not increase significantly. This leads to the compensation of ageing in the operated pumping well by reducing the flow rate. For $\dot{Q} < 0$ *and* $\dot{\alpha} \approx 0$ the following can be specified:

$$\Delta\eta_{cmp} = -(\dot{\alpha}Q + \alpha\dot{Q})\cdot\eta_1^2\cdot\Delta t_{cmp}\Delta\eta_{cmp} = -\alpha\dot{Q}\cdot\eta_1^2\cdot\Delta t_{cmp} \tag{14}$$

Obviously, in the above case the inequality $\Delta\eta_{cmp} > 0$ over time will always be satisfied.

It should be pointed out here that a well with an initial efficiency of $\eta_0$ after the expiry of the period of ageing $\Delta t_{age}$ reaches a state with the hydraulic efficiency of $\eta_1$. This state represents the initial state for the later compensation of the well loss value. After the compensation time $\Delta t_{cmp}$ the well reaches an efficiency of $\eta$. The number of iterations depends on the manner of transition between the extreme states of the well desired by the user.

*3.2. Second-Order Approximation*

In this case, it was assumed that higher-order terms, starting from the third order, are negligible. Therefore, the power series expansion of the well efficiency function is as follows:

$$\Delta\eta(t) \approx \dot{\eta}(t_0)\cdot\Delta t + \frac{1}{2}\ddot{\eta}(t_0)\cdot\Delta t^2 \tag{15}$$

while ensuring that the time derivatives at the moment of time $t = t_0$ are equal to:

$$\begin{aligned} \dot{\eta}(t_0) &= -(\dot{\alpha}Q + \alpha\dot{Q})\cdot\eta_0^2 \\ \ddot{\eta}(t) &\approx \tfrac{2}{\eta_0}[\dot{\eta}(t_0)]^2 = 2(\dot{\alpha}Q + \alpha\dot{Q})^2\cdot\eta_0^3 \end{aligned} \tag{16}$$

Therefore, the relationship for the well efficiency change will be equal to:

$$\begin{aligned} \Delta\eta(t) &\approx -(\dot{\alpha}Q + \alpha\dot{Q})\cdot\eta_0^2\cdot\Delta t + \tfrac{1}{2}2(\dot{\alpha}Q + \alpha\dot{Q})^2\cdot\eta_0^3\cdot\Delta t^2 \\ \Delta\eta(t) &\approx -(\dot{\alpha}Q + \alpha\dot{Q})\cdot\eta_0^2\cdot\Delta t + (\dot{\alpha}Q + \alpha\dot{Q})^2\cdot\eta_0^3\cdot\Delta t^2 \\ \Delta\eta(t) &\approx (\dot{\alpha}Q + \alpha\dot{Q})\cdot\eta_0^2\cdot\Delta t[(\dot{\alpha}Q + \alpha\dot{Q})\cdot\eta_0\cdot\Delta t - 1] \end{aligned} \tag{17}$$

The first element of the difference provided in square brackets could be significantly smaller than one. This is possible because the time derivative of the product $\dot{\alpha}Q + \alpha\dot{Q} = \frac{d}{dt}(\alpha Q)$ for a short time (where $\Delta t \to 0$) is low or slow-variable. Therefore, for $\frac{d}{dt}(\alpha Q) \approx 0$ or $\alpha Q \approx const$ and $\Delta t \to 0$ an accurate expression for well efficiency change can be obtained at first-order approximation.

*Case 1—Well Ageing*

Well ageing occurs with an invariable flow rate and an increase in the drawdown observed in the pumping well through a significant increase in well resistances and insignificant increase in aquifer resistances.

For $\dot{Q} \approx 0$ *and* $\dot{\alpha} > 0$ the simplified form of relationship (17) applies:

$$\Delta\eta_{age} = -\dot{\alpha} \cdot Q \cdot \eta_0^2 \cdot \Delta t_{age}[1 - \dot{\alpha} \cdot Q \cdot \eta_0 \cdot \Delta t_{age}] \tag{18}$$

The above equation will be satisfied for the loss of hydraulic efficiency of a well ($\Delta\eta_{age} < 0$) if the passage of time is expressed in a limited range, given as the strict inequality $\Delta t_{age} < (\dot{\alpha} \cdot Q \cdot \eta_0)^{-1}$.

*Case 2—Well Ageing Compensation*

Flow rate decreases in an appropriately short time, while well resistances do not significantly increase over this time. For $\dot{Q} < 0$ *and* $\dot{\alpha} \approx 0$ it will be:

$$\Delta\eta_{cmp} = -\alpha \dot{Q} \cdot \eta_1^2 \cdot \Delta t_{cmp}[1 - \alpha \dot{Q} \cdot \eta_1 \cdot \Delta t_{cmp}] \tag{19}$$

As can be easily noticed, expression (19) describes inequality $\Delta\eta_{cmp} > 0$ with time and decreasing flow rate in the well over time. In other words, it is possible to recover at least a part of the hydraulic efficiency lost. As before, after time $\Delta t_{age}$, well efficiency decreased from $\eta_0$ to $\eta_1$, and it constitutes the initial value for the compensation of the well loss through a reduction in the flow rate of the borehole at time $\Delta t_{cmp}$. The number of iterations depends on the manner of transition between the extreme states of the well desired by the user.

## 4. Discussion

The archived results can be used in practice to compensate for well ageing. Taking into account that equations are difficult to use in practice, they can be expressed in a more practical way. Particular emphasis is put on the first order-approximation. For this purpose, characteristic indicators for full compensation of ageing will be presented below. In addition, expressions defining the necessary intervals to be taken in order to carry out corrective action are presented. The usage of those indicators for the selected example will be presented in the following section.

*4.1. First Order Approximation*

4.1.1. Compensation of Well Ageing

It follows from case 1 that efficiency $\eta_1$ is:

$$\eta_1 = \eta_0 - \dot{\alpha} \cdot Q \cdot \eta_0^2 \cdot \Delta t_{age} \tag{20}$$

while obviously always $\Delta\eta_{age} = \eta_1 - \eta_0 < 0$. Case 2 indicates that the final efficiency $\eta$ value is:

$$\eta = \eta_1 - \alpha \dot{Q} \cdot \eta_1^2 \cdot \Delta t_{cmp} \tag{21}$$

while this time always $\Delta\eta_{cmp} = \eta - \eta_1 > 0$.

The insertion of the expression for efficiency $\eta_1$ results in a change in efficiency as a result of compensation as:

$$\begin{aligned}
\Delta\eta_{cmp} &= -\alpha \dot{Q} \cdot \eta_1^2 \cdot \Delta t_{cmp} \\
\Delta\eta_{cmp} &= -\alpha \dot{Q} \cdot (\eta_0 - \dot{\alpha} \cdot Q \cdot \eta_0^2 \cdot \Delta t_{age})^2 \cdot \Delta t_{cmp} \\
\Delta\eta_{cmp} &= -\alpha \dot{Q} \cdot \eta_0^2 \cdot \Delta t_{cmp}(1 - \dot{\alpha} \cdot Q \cdot \eta_0 \cdot \Delta t_{age})^2
\end{aligned} \tag{22}$$

4.1.2. Full Compensation of Well Ageing

It is readily visible that the total change of well efficiency equals:

$$\Delta\eta = \eta - \eta_0 = (\eta - \eta_1) + (\eta_1 - \eta_0)\Delta\eta = \Delta\eta_{cmp} + \Delta\eta_{age} \tag{23}$$

the sign of the total change (difference) in efficiency is determined by the size of the individual elements.

Assuming that time moves towards moment $t_0$, i.e., $t \to t_0$, so the time range becomes infinitesimal ($\Delta t_i \to 0$) then the decrease in well efficiency as a result of ageing also becomes infinitesimal ($\Delta \eta_{age} \to 0$ or $\eta_1 \to \eta_0$). Furthermore, an increase in well efficiency as a result of compensating the well loss will also be infinitesimal ($\Delta \eta_{cmp} \to 0$ or $\eta \to \eta_1$). As for the rate of each of the above changes, it will be given using the "0/0" indeterminate form. However, it follows from the principle of transitivity of implication that $\eta \to \eta_0$ also occurs, therefore in general the transition of the function to the limit $\Delta \eta \to 0$ where $\Delta t \to 0$ must occur. This means that the equality of function limits will be satisfied.

$$\lim_{\Delta t \to 0} \Delta \eta = \lim_{\Delta t \to 0} \Delta \eta_{age} = \lim_{\Delta t \to 0} \Delta \eta_{cmp} = 0 \tag{24}$$

Equations (23) and (24) provide knowledge about the possibility of the following relationship:

$$\Delta \eta = \Delta \eta_{age} + \Delta \eta_{cmp} = 0 \tag{25}$$

It is always satisfied not only for initial moments of time, but also independently of time, which occurs with a full compensation of the decrease in well efficiency.

The insertion of expressions (13) and (14) into (25) results in:

$$\dot{\alpha} Q \cdot \eta_0^2 \cdot \Delta t_{age} + \alpha \dot{Q} \cdot \eta_1^2 \cdot \Delta t_{cmp} = 0 \tag{26}$$

Applying Leibniz's rules to the differential calculus for function composition (the chain rule), it can be specified that if the proposition $\frac{df}{dx} = \frac{\frac{df}{dt}}{\frac{dx}{dt}} \Rightarrow \frac{df}{dx} \frac{dx}{dt} = \frac{df}{dt}$ is true, then the time from derivatives in expression (26) can be easily erased to obtain a first-order homogenous differential equation depending on the variable parameters $\alpha$ and $Q$:

$$\frac{d\alpha}{dQ} \cdot \eta_0^2 \cdot \Delta t_{age} + \frac{\alpha}{Q} \cdot \eta_1^2 \cdot \Delta t_{cmp} = 0 \tag{27}$$

The separation of variables makes it possible to express the formula in the following form:

$$\begin{aligned} -\frac{d\alpha}{\alpha} &= \frac{dQ}{Q} \cdot \left(\frac{\eta_1}{\eta_0}\right)^2 \left(\frac{\Delta t_{cmp}}{\Delta t_{age}}\right) \\ -\frac{d\alpha}{\alpha} &= \beta \frac{dQ}{Q} \end{aligned} \tag{28}$$

The integration of relationship (28) with the constant of integration written as $\ln\gamma_0 = const$ and assuming for further simplicity constancy of coefficient $\beta$ will result:

$$\begin{aligned} -\int \frac{d\alpha}{\alpha} &= \int \beta \frac{dQ}{Q} = \beta \int \frac{dQ}{Q} \\ -\ln \alpha &= \beta \ln Q + const \\ \ln \alpha^{-1} &= \ln Q^\beta + \ln \gamma_0 = \ln \left(\gamma_0 Q^\beta\right) \end{aligned} \tag{29}$$

After introducing coefficient $\delta = \gamma_0^{-1}$ the above expression can be written in an antilogarithm (exponential) formula:

$$\begin{aligned} \alpha^{-1} &= \gamma_0 Q^\beta \\ \gamma_0^{-1} &= \delta = \alpha Q^\beta \end{aligned} \tag{30}$$

where exponent $\beta$ of the power is equal to:

$$\beta = \left(\frac{\eta_1}{\eta_0}\right)^2 \left(\frac{\Delta t_{cmp}}{\Delta t_{age}}\right) \tag{31}$$

Exponent $\beta$ can assume different values, but some characteristic values can be listed. The following is a discussion of cases where: $\beta = 0$ and $\beta = 1$.

When $\beta = 0$, it should be expected that there is a strict inequality for the hydraulic efficiencies of the well at specific moments of time, i.e., $\eta_1 < \eta_0$. Therefore, their quotient is $\frac{\eta_1}{\eta_0} < 1$ and the square of quotient is $\left(\frac{\eta_1}{\eta_0}\right)^2 \ll 1$. Furthermore, it can be assumed that the time available for compensating well loss is shorter than the well ageing time. It is therefore realistic to assume that the relationship $\Delta t_{cmp} < \Delta t_{age}$ will be true each time. This means that the exponent in Equation (31) will be equal to $\beta << 1$. In the context of mathematical analysis, it proceeds in its limit to the value of $\beta \to 0$. Hence, expression (30) must assume the form:

$$\lim_{\beta \to 0} \delta = \lim_{\beta \to 0} (\alpha Q^\beta) = \lim_{\beta \to 0} (\alpha Q^0) = \lim_{\beta \to 0} \alpha = \alpha$$
$$\lim_{\beta \to 0} \delta = \alpha = \frac{C}{B} = \delta_0 \tag{32}$$

Therefore, when the user of a water supply well is determined to achieve full compensation of the well loss in a wellbore used over a span of many years, then a certain stable state between the resistances of the aquifer and the resistances of the well should be established for the water flow in the aquifer. Indeed, as provided by Equation (32):

$$C = \alpha \cdot B \quad or \quad C = \delta_0 \cdot B \quad when \quad \beta \approx 0 \tag{33}$$

The delta ($\delta_0$) parameter does depend on the property of the area directly adjacent to the functional part of the well screen and the permeability of the rock-soil medium conducting the fluid, as well as the type of the flowing reservoir medium.

Taking into account Equations (1) and (33), the hydraulic efficiency of the well can be equal to:

$$\eta = \frac{1}{1 + \alpha Q} = \frac{1}{1 + \delta_0 Q} \quad when \quad \beta \approx 0 \tag{34}$$

In other words, the ageing and compensation time ranges assumed for the well are similar in value ($\Delta t_{age} \approx \Delta t_{cmp}$) and the well efficiencies compared to the limits of the structure's ageing period range are almost identical ($\eta_0 \approx \eta_1$), then the exponent from Equation (31) equals $\beta \approx 1$. Obviously, this case will occur for initial moments—in situations where $\Delta t_{age} \to 0$ and $\Delta t_{cmp} \to 0$, i.e., where $\eta_1 \to \eta_0$ (or $\Delta \eta_{age} \to 0$). Therefore, the limit:

$$\lim_{\beta \to 1} \delta = \lim_{\beta \to 1} (\alpha Q^\beta) = \lim_{\beta \to 1} (\alpha Q^1) = \lim_{\beta \to 1} \alpha Q = \alpha Q$$
$$\lim_{\beta \to 1} \delta = \alpha \cdot Q = \frac{C}{B} \cdot Q = \delta_1 \tag{35}$$

Then:

$$\delta_1 = \alpha \cdot Q = \frac{C}{B} \cdot Q = \frac{CQ^2}{BQ} \quad or \quad \delta_1 = \frac{s_2}{s_1} \quad when \quad \beta \approx 1 \tag{36}$$

Parameter $\delta_1$ for the initial moment under the conditions described above defines the mutual relationships between well loss and aquifer depression (or generally for a pool of any reservoir medium).

The hydraulic efficiency of a water supply well in accordance with the initial Jacob's formula, Equation (1), is then equal to:

$$\eta = \frac{s_1}{s_1 + s_2} = \frac{s_1}{s_1 + \delta_1 \cdot s_1} = \frac{1}{1 + \delta_1} \quad when \quad \beta \approx 1 \tag{37}$$

A table listing was prepared to summarize the discussion on the full compensation of well loss in the operated pumping well (Table 1).

**Table 1.** A listing of the calculation results for the full compensation of ageing-related well efficiency decrease, relationship of the type $\delta = \alpha \cdot Q^\beta$ for $\Delta\eta = \Delta\eta_{age} + \Delta\eta_{cmp} = 0$.

| $\beta$ | $\delta_\beta$ | $\eta$ | Note |
|---|---|---|---|
| 0 | $Q_0^{-1} = \alpha = C/B$ | $1/(1 + \delta_0 \cdot Q)$ | Full compensation—long period of well operation |
| 1 | $s_2/s_1 = \alpha \cdot Q$ | $1/(1 + \delta_1)$ | Full compensation—initial moments of well operation |

### 4.1.3. Compensation Time

From Equations (30) and (31) the time required for the full compensation of hydraulic efficiency of the water supply well can be calculated. The following expression can be written:

$$\beta = \frac{\ln \frac{\delta}{\alpha}}{\ln Q} = \left(\frac{\eta_1}{\eta_0}\right)^2 \left(\frac{\Delta t_{cmp}}{\Delta t_{age}}\right) \tag{38}$$

Therefore, compensation time can be calculated using the formula:

$$\Delta t_{cmp} = \frac{\ln \frac{\delta}{\alpha}}{\ln Q} \cdot \left(\frac{\eta_0}{\eta_1}\right)^2 \cdot \Delta t_{age} \tag{39}$$

### *4.2. Second Order Approximation*

Omitting the second element of the factor provided in square brackets in Equations (18) and (19) makes it possible to obtain a first-order approximation in such special cases. Writing down further equations would blur clarity due to complicated notation. At this stage Equations (18) and (19) provide information which allows taking action aimed at improving the hydraulic efficiency of the well.

The above discussion clearly demonstrates that periodical step drawdown tests play a fundamental role in the assessment of hydraulic state and also the potential actions aimed at extending the period of intake operation at a high level of hydraulic efficiency. However, the increase in well loss could be observed during regular or occasional inspections of operating parameters, i.e., drawdown and flow rate. In practice, compensation of the loss of hydraulic efficiency is achieved by methods such as valve throttling, switching the type series of pump units or, alternatively, regulating the rotational speed of the pump impeller.

Under conditions of a full compensation of wellbore ageing, parameter $\delta_0$ is numerically related to the flow of the reservoir medium for which the flow rate of a water supply well $Q_0 = (C/B)^{-1}$ results in the hydraulic efficiency of the well equal to 0.5 (50%), i.e., where well loss ($s_2$) is equal to aquifer loss ($s_1$). An increase in the value of parameter $\delta_0$ means that the well reaches the same level of hydraulic efficiency with a lower volumetric flow rate. This is combined with a progressive increase in the resistance coefficient value of the well in turbulent flow at the well-screen near zone in relation to the resistance coefficient of laminar flow in the aquifer. In other words, parameter $\delta_0$ may constitute a reliable indicator of well condition.

For a sufficiently long period of operation of the well the exponent is $\beta = 0$. Then for factor $\delta = \delta_0 = \alpha = C/B$ the time required to compensate well loss $\Delta t_{cmp} = 0$, i.e., in the conditions of well operation, is $\Delta t_{cmp} \ll \Delta t_{age}$. This means that for the long and intensive operation of water supply wells, flow rate reductions and compensation of hydraulic efficiency loss must not be delayed. For exponent $\beta = 1$, i.e., at the initial stage of the lifespan of a water supply well, factor $\delta = \delta_1 = s_2/s_1 = \alpha Q$ ensures compensation time at a level equal to the time of ageing, i.e., operation of the structure—$\Delta t_{cmp} \approx \Delta t_{age}$, provided that the efficiencies assume similar values ($\eta_0 \approx \eta_1$). In other words, the time available for taking corrective actions for the well becomes extended and is similar in its order of magnitude to the time of operation of the wellbore.

Furthermore, the higher the initial efficiency of the well (in relation to that expected after the period of operation of the well), the later it will be possible to begin compensating the loss of hydraulic efficiency by regulation through discharge, e.g., in an active or passive manner. On the other hand, excessively intensive pumping from the well will result in reducing the time available for compensating the effect of well loss. The well loss should increase with the clogging of the well-screen zone, manifested in the increase in the resistance coefficient value *C*, i.e., with the turbulent flow in the aquifer. Ultimately, this manifests itself in a decrease in the hydraulic efficiency curve of the well. Moreover, greater curvature of the graph is observed.

*4.3. Case Study*

Figures 2 and 3 present an example of using the solution presented above. Initially, Well 1 (Figure 2) operated at the flow rate of $12.5 \times 10^{-3}$ m$^3$/s, however parameter $\alpha^{-1} = B/C = Q_0 = 37.4 \times 10^{-3}$ m$^3$/s. After 10 years of uninterrupted operation there was an unforeseen failure connected with the exposure of the pumping unit. A pumping test demonstrated a well loss of up to 80% of total drawdown. To protect the new pump unit, the pumping rate was limited to $5.6 \times 10^{-3}$ m$^3$/s, which corresponded to the hydraulic efficiency of the well of 30%. The conducted rehabilitation procedures failed to bring the expected results. The tests demonstrated that the final low efficiency of the well was caused by the relatively low initial well efficiency and the lack of efficiency compensation during operation. It has led to gravel-pack clogging and particulate damage. Finally, the value $Q_0$ decreased to $2.4 \times 10^{-3}$ m$^3$/s, which corresponds to an efficiency of 50%.

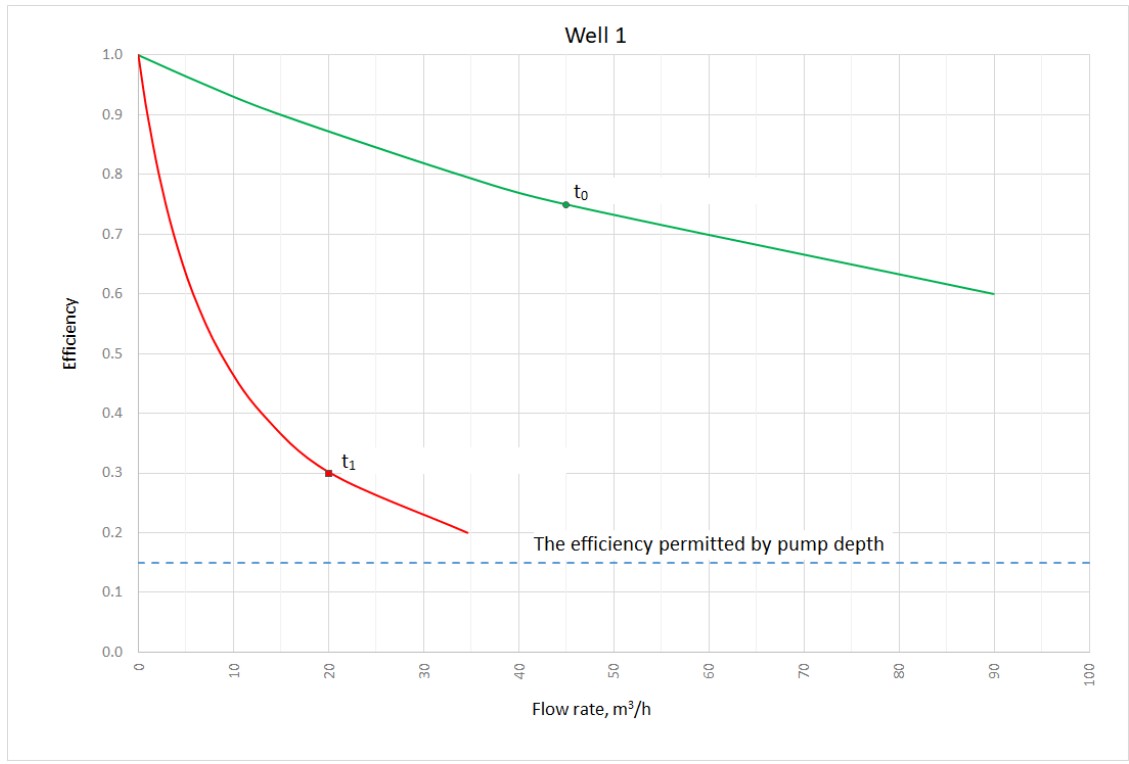

**Figure 2.** The hydraulic efficiency of the well without flow rate compensation: $t_0$—at the beginning of operation time; $t_1$—at the end of operation time.

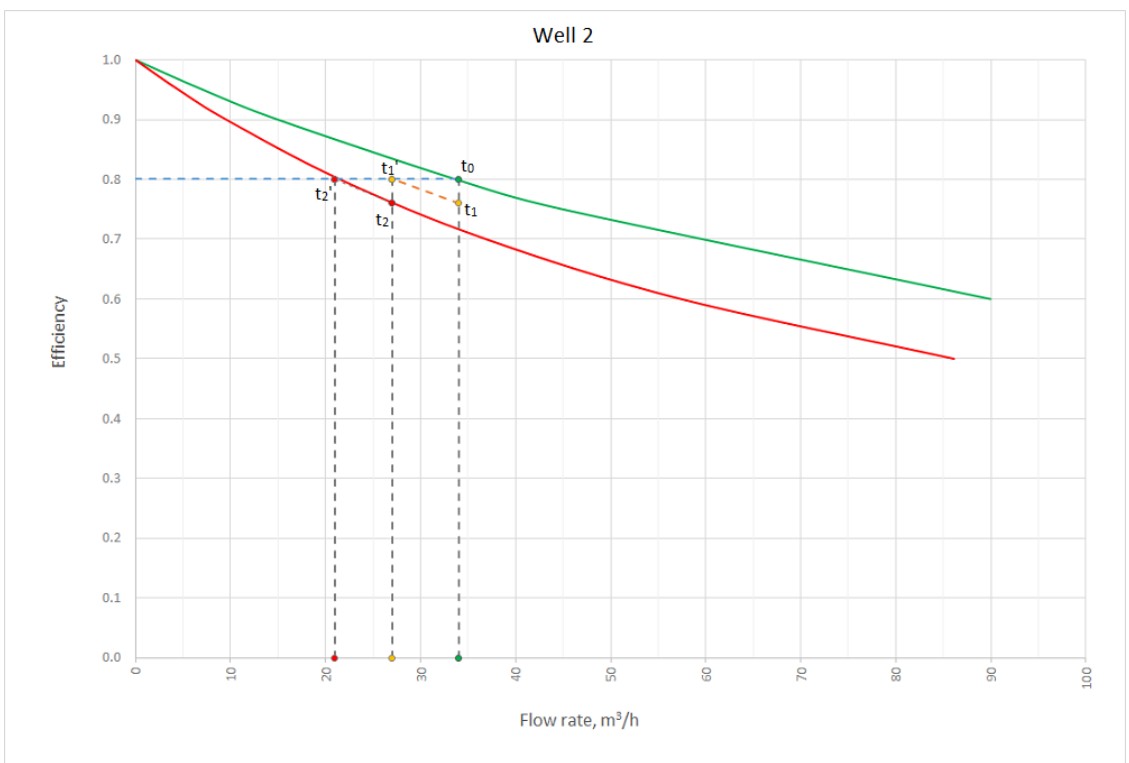

**Figure 3.** The hydraulic efficiency of the well with flow rate compensation: $t_0$—the working point at the beginning of operation; $t_1$—the working point before first flow rate compensation; $t_{1'}$—the working point after first flow rate compensation; $t_2$—the working point before second flow rate compensation; $t_{2'}$—the working point after second flow rate compensation; $\eta_0$—the origin efficiency curve; $\eta_2$—the current efficiency curve.

To reduce operating costs, the company conducting groundwater extraction constructed a replacement well. The methods presented in this article were used during its operation (Figure 3). Initially, the maintenance of an efficiency of 80% was established, which corresponded to the flow rate of $9.4 \times 10^{-3}$ m$^3$/s, however parameter $Q_0 = 37.1 \times 10^{-3}$ m$^3$/s. After a further eight years the reduced flow rate was $5.8 \times 10^{-3}$ m$^3$/s with a constant efficiency. In this time, the value $Q_0$ decreased to $23.9 \times 10^{-3}$ m$^3$/s, which corresponds to an efficiency of 50%.

It should be noted that flow rate was significantly reduced for both wells. Currently, Well 1 is used occasionally due to high operating costs, while the costs of water extraction from Well 2 are approximately 65% lower. In addition, Well 2 operates at an efficiency which guarantees its reliable operation in the long term. The efficiency values, which are marked green and red in Figures 2 and 3, were confirmed by step drawdown tests. In practice, efficiency compensation in the second case was achieved by valve throttling at the beginning of operation and replacing the pump unit type series at appropriate time [16].

## 5. Conclusions

The solution presented in this paper involves ageing compensation and maintaining as high a hydraulic efficiency as possible. It concerns water supply wells in which well-loss has a minor share in the drawdown generated by water pumping. So, the presented methodology is valid to both new and old wells in good hydraulic condition, where hydrogeological conditions do not generally change over time. The fundamental meaning in presented method has parameter $\alpha$. This parameter is a relation $C$ to $B$, and may constitute a reliable well condition index which is relevant to 0.5 (50%) efficiency of the well.

Unlike "sustainable yield", concept of efficiency, compensation is not constant-rate or constant-head method. The "sustainable efficiency" concept requires regulation of both flow rate and drawdown. In relation to aquifer recharge fluctuations, operational parameters could be temporarily increased when water larger resources are available. However, in general, operating parameters of the well are reduced over time. This approach allows for the maintenance of a constant efficiency of the well, as long as the water output is sufficient to cover the water demand. Then, rehabilitation of the well is needed. Taking into consideration compensation time, it should be expressed that regulation time is dependent on current efficiency of the well. When the initial efficiency is relatively high, the compensation can be postponed or delayed. Alternatively, when the initial efficiency is low the time to reducing operational parameters is shortened.

The methodology was applied in a groundwater extraction company for new Well 2, drilled as a substitute of a Well 1, where efficiency was dramatically reduced over 10 years of constant water extraction. The periodical efficiency reduction of Well 2 during similar times of operation forced both the pumping rate and drawdown reduction. Currently, it has resulted in a relatively low cost of water extraction. Furthermore, the relation *B* to *C* still remains on a relatively high-level value. This indicates that the well-loss is quite small and the particulate damage in the near-well zone was minimized. The upcoming rehabilitation of Well 2 will probably be much more effective in this case. It can also be noticed that theoretical considerations presented in this paper were verified by step-drawdown tests at the beginning and also at the end of the research period. So, the method can be applied as a practical solution in well management to prevent inefficiencies.

**Author Contributions:** K.P. and K.K.-O. conceived of the presented idea. K.G. developed the theory and performed the computations. K.P. verified the analytical methods, performed the measurements and field test data calculations. K.K.-O. contributed to the interpretation of the results. All authors discussed the results and contributed to the final manuscript.

**Funding:** This research received no external funding. The APC was funded by AGH University of Science and Technology, Krakow 30-059, Poland.

**Acknowledgments:** The authors are grateful for the constructive comments made by anonymous reviewers which helped to improve the manuscript.

**Conflicts of Interest:** The authors declare no conflict of interest.

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
