# Peer review of "The Dynamics of Water Wells Efficiency Reduction and Ageing Process Compensation"

_water, doi:10.3390/w11010117_

Round 1

Reviewer 1 Report

Review of

The dynamics of water wells efficiency reduction and aging process compensation

by Polak et al.

This manuscript presents a series of mathematical analyses about well aging.  The authors claimed that the work makes it possible to determine when necessary adjustments of the operating parameters should be performed. The manuscript presents some efforts on calculations, I recommend for publication with minor revision.

Detailed comments can be found below.

Comments

The authors gave some recommendations for operating parameters in field. Please provide some field evidence that the recommendations are applicable (e.g., compare with reported field data).

Make the word “ageing” consistent (e.g., line 4).

Author Response

Response to Reviewer 1 Comments

1.     The authors gave some recommendations for operating parameters in field. Please provide some field evidence that the recommendations are applicable (e.g., compare with reported field data).

In the new version of manuscript we provided the field evidence example on Figure 2 and Figure 3 . This is graphical presentation of field test data. The description of results are also included in new CASE STUDY chapter. 

2.     Make the word “ageing” consistent (e.g., line 4).

This line was not necessary and its deleted. 

I would like to thank for your kind revision and recommendation.

Reviewer 2 Report

[Water] Manuscript ID: water-405320 – Review

General comments

This paper by Krzysztof Polak, Kamil Górecki and Karolina Kaznowska-Opala is focused on the degradation process of water wells as a variable depending on the operation time, the well loss and the flow rate. The overall aim of the paper is to determine the moment of ageing compensations of the degradation processes of the water wells. Specifically, the authors highlight the importance of the correlation between hydraulic resistances in an aquifer and in the engineering structure. The aim of the analysis is to contribute to the tracking of the changes in the ageing process occurring in the engineering structures. The paper should be improved in terms of clarity and quality. An improved revised manuscript would address the following major comments.

Major Comments

The manuscript      needs improvement regarding the structure of the sentences, paragraphs and      most importantly sections of the paper (e.g. Introduction, Materials and      methods, Results, Discussion). The authors do not fully follow the      structure of the manuscript template as this is provided from the journal.

The Introduction of      this paper consists of only two paragraphs. The authors should discuss in      more detail what has been done before by previous researchers. Please do      not add a big number of references for only a single sentence in your      paper (e.g. Lines: 40, 44). Instead please try to discuss the work of      these researchers, which is relevant to your study. Also, in the Materials      and Methods part, readers can only read equations and parameters; very few      complete sentences are included.

The paper is      missing an important amount of references. Citations in the introduction      referring to previous work are only superficially mentioned without any      further information or relevance to the rest of the paper. The Introduction,      Materials and Methods, Results and Discussion sections should be re-structured      so that more information on previous research is available and gaps in      knowledge are clearly described supporting the aim of this paper and      proving its novelty.

The authors      should be more explanatory when starting a new section in their paper. The      readers cannot be introduced to the problem discussed as the paper is      presented at the moment. This is very important for the Introduction, Materials      and Methods, and Results and Discussion parts. These parts need      significant improvement. However, the Abstract and Conclusions parts are      well presented by the authors.

Please try to form      your Conclusions without the presence of equations and parameters. This way it will be easier to read. There is plenty of space at      the Discussion part to explain in detail these parameters (where is the      Discussion part in your paper?). Please include only your interesting and      valuable conclusions in this part of your paper. Union the last three      paragraphs of your conclusions as they are too short to be in the paper as      separate paragraphs.

Author Response

Dear Reviewer,

I would like to thank you for your helpful review. 

Our response you will find in attached PDF file.

Kind regards,

Krzysztof Polak

Round 2

Reviewer 2 Report

[Water] Manuscript ID: water-405320 – Review 2

General comments

The authors improved a lot the paper in terms of clarity and quality from the previous review. However, an improved revised manuscript would address the following major comments that are very important.

Major Comments

The authors      improved the Introduction section of this paper that previously consisted      of only 2 paragraphs. The authors should start in a more general way the      Introduction. The authors should also fix the references of the paper; first      reference starts with number 25. The second paragraph of the Introduction      does not need to be separated from the previous paragraph. The authors      should not create such short paragraphs or refer to Figures in the      Introduction. The authors should break paragraph 4 into 2 paragraphs as it      is too long. Please cite the work of other researchers based on the      instructions of Water journal. The 2 paragraphs that start with line 107      and 114, respectively, can be joined.

The Materials and      Methods part is still too short. The authors should explain in more details      their 2 assumptions on which their study is based. Again, the authors      should avoid too short paragraphs.

The paper is still      missing again references. The authors need to add more references in their      paper. Most references are way too old. The authors      should try to find more up to date references that are relevant to their      work.

Again, the      authors should be more explanatory when starting a new section in their      paper. The authors did not improve that a lot throughout their paper.      Every single paragraph throughout the paper should start with a small      “introduction” so that the readers can follow what the authors try to      address.  For example, the Results      and the Discussion should not start like that.

Please check your      Figures if they meet the journal’s requirements.

The Conclusions section      is too long. The authors can include some of this part of      the Conclusions back to the Discussion. As mentioned before, please include      only your interesting and valuable conclusions in this section of your      paper. Notice your paragraph structure at this section.

The authors should check for spelling mistakes throughout the      paper 

The authors should reply to the Reviewer’s comments; answering each      of these points one by one instead of the Reviewer to try to find the      corrections made by the authors throughout the whole paper. Please add a      short comment close to each change when needed.

Author Response

Response to Reviewer 2 Comments

The reviewer general comments:

The authors improved a lot the paper in terms of clarity and quality from the previous review. However, an improved revised manuscript would address the following major comments that are very important.

Major Comments:

Point 1: The authors improved the Introduction section of this paper that previously consisted of only 2 paragraphs. The authors should start in a more general way the Introduction. The authors should also fix the references of the paper; first reference starts with number 25. The second paragraph of the Introduction does not need to be separated from the previous paragraph. The authors should not create such short paragraphs or refer to Figures in the Introduction. The authors should break paragraph 4 into 2 paragraphs as it is too long. Please cite the work of other researchers based on the instructions of Water journal. The 2 paragraphs that start with line 107 and 114, respectively, can be joined.

Response 1: More general start of Introduction was included (lines 33-38). References are numbered in order of appearance in the text. First and second paragraph are jointed in one (line 33-44). Previously, paragraphs (1 and 2) was braked in different issues. It was braked once again according to review 2. 11 new articles was cited in new version of manuscript according to Instructions of Water journal. Previous paragraphs lines 107-114 are jointed (new lines: 112-125).

Point 2. The Materials and Methods part is still too short. The authors should explain in more details their 2 assumptions on which their study is based. Again, the authors should avoid too short paragraphs.

Response 2: Materials and Methods is much longer at this moment. Previous summary of Introduction section now is used to explain fundamentals of our own considerations presented in Materials and Methods section (lines: 187-201). Moreover, the example taken from referenced paper was included in "little intro" to set out presumptions of our own theoretical considerations (lines: 163-186). 2 assumptions are explained in more details in lines: 202-221.

Point 3: The paper is still missing again references. The authors need to add more references in their paper. Most references are way too old. The authors should try to find more up to date references that are relevant to their work.

Response 3. The new 11 references dated on this century was cited in 3thd manuscript. Half of all references are dated on last 20 years. The references dated before 1990 are fundamental in the research area and can't be ignored. We don’t know the references directly relevant to dynamics of water wells ageing process. However, paper 25, 39 and 40 are related indirectly to the issue.

Point 4. Again, the authors should be more explanatory when starting a new section in their paper. The authors did not improve that a lot throughout their paper. Every single paragraph throughout the paper should start with a small “introduction” so that the readers can follow what the authors try to address.  For example, the Results and the Discussion should not start like that.

Response 4: As it mentioned above, the explanation/intro to Material and Methods section was included in new manuscript (lines 162-221). Results and Discussion also contains “small intro” (lines: 243-250 and 312-317). 

Point 5: Please check your Figures if they meet the journal’s .

Respons 5: The figure requirements have been re-checked. In my opinion figures corespond to journal  requirements.

Point 6: The Conclusions section is too long. The authors can include some of this part of the Conclusions back to the Discussion. As mentioned before, please include only your interesting and valuable conclusions in this section of your paper. Notice your paragraph structure at this section. 

Response 6: Previous conclusions returned to the discussion part (lines 395-426). A new section Conclusions has been included in the manuscript (lines 464-490). Nowadays, it is much more general. 

Point 7: The authors should check for spelling mistakes throughout the paper.

Response 7: All text had been check by 2 independent, professional English speakers.

Point 8: The authors should reply to the Reviewer’s comments; answering each of these points one by one instead of the Reviewer to try to find the corrections made by the authors throughout the whole paper. Please add a short comment close to each change when needed.

Response 8: All corrections are generally visible in the attached file. However, I am aware of the fact that it is not visible enough in the details. So all corrections are listed above one by one (line number). I hope that the above explanations are adequate and comprehensive.  

Once again, I would like to thank you for the constructive comments which helped to improve the manuscript.

Best regards and happy new year,

Krzysztof Polak

Round 3

Reviewer 2 Report

[Water] Manuscript ID: water-405320 – Review 3

General comments

The authors: Krzysztof Polak, Kamil Górecki and Karolina Kaznowska-Opala made a significant effort to answer all major suggestions of the Reviewer based on the previous Review. There are specific recommendations given from the Reviewer to the authors to make them understand what remains to be improved. Please read the following Minor corrections.

Minor Comments

Please      delete blank lines 199, 448 and 494 in the paper.

The      authors should explain a bit more Figure 1 capture with the different      losses illustrated in the figure.

Paragraphs      starting with line 213 and 217, respectively, can be joined. There is no      need to be separated because they present the 2 different assumptions.

Line      262, please add “,” before the word which.

Line      110, please remove the word “etc” unless, you know the other effects as      well. In this case, you should include all the other effects that you      know.

Why      the authors keep on having short paragraphs? Please improve that in the      Results section as it is not good for the readers. For example: lines      306-310.

Line      479: Please explain this sentence. What the authors mean by “moment is      moved into the future”.

Author Response

[Water] Manuscript ID: water-405320 – Review 3

Response to Review 2 Comments

General comments

The authors: Krzysztof Polak, Kamil Górecki and Karolina Kaznowska-Opala made a significant effort to answer all major suggestions of the Reviewer based on the previous Review. There are specific recommendations given from the Reviewer to the authors to make them understand what remains to be improved. Please read the following Minor corrections.

Minor Comments

Comment 1: Please delete blank lines 199, 448 and 494 in the paper.

Response 1. Blank lines have been removed.

Comment 2:The authors should explain a bit more Figure 1 capture with the different losses illustrated in the figure.

Response 2: Figure 1 has been corrected. The descriptions on the figure explain the components of drawdown in the well. The description under the figure has been expanded and explains the essence of turbulence around the well. 

Comment 3:Paragraphs starting with line 213 and 217, respectively, can be joined. There is no need to be separated because they present the 2 different assumptions. 

Response 3: We tried to do it as suggested by the reviewer. We are convinced that the previous division is much clearer. So, paragraphs beginning with lines 213 and 217 are now slightly improved. However, we would like to keep the previous form.

Comment 4:Line 262, please add “,” before the word which.

Response 4: Line 262 has been corrected.

Comment 5: Line 110, please remove the word “etc” unless, you know the other effects as well. In this case, you should include all the other effects that you know.

Response 5: We do not know the other effects (which does not exclude that they exist). The word "etc" has been removed. 

Comment 6: Why the authors keep on having short paragraphs? Please improve that in the Results section as it is not good for the readers. For example: lines 306-310.

Response 6: With paragraphs we affect the different meanings of the text. As suggested by the reviewer lines 306-310 have been jointed (new lines: 303-308).

Comment 7: Line 479: Please explain this sentence. What the authors mean by “moment is moved into the future”?

Response 7: The compensation time depends on the initial hydraulic condition. The sentence in line 479 is a consequence of the previous one. It has been changed (new line: 476). We hope, it is correct at present form.

I would like to notice that line 424 has been supplemented by one sentence. We hope that our explanations and improvements will be accepted.

Thank you for your kind cooperation.

Best regards,

Krzysztof Polak
